# Transcriptomic Analysis of Hormone Signal Transduction, Carbohydrate Metabolism, Heat Shock Proteins, and SCF Complexes before and after Fertilization of Korean Pine Ovules

**DOI:** 10.3390/ijms24076570

**Published:** 2023-03-31

**Authors:** Xiaoqian Yu, Xueqing Liu, Yuanxing Wang, Yue Zhang, Hailong Shen, Ling Yang

**Affiliations:** 1State Key Laboratory of Tree Genetics and Breeding, School of Forestry, Northeast Forestry University, Harbin 150040, China; yxq384811@163.com (X.Y.); lxq19846151691@163.com (X.L.);; 2Jilin Provincial Academy of Forestry Sciences, Changchun 130033, China; 3State Forestry and Grassland Administration Engineering Technology Research Center of Korean Pine, Harbin 150040, China

**Keywords:** Korean pine, ovule development, anatomical structure observation, fertilization observation, RNA-seq, qRT-PCR

## Abstract

The fertilization process is a critical step in plant reproduction. However, the mechanism of action and mode of regulation of the fertilization process in gymnosperms remain unclear. In this study, we investigated the molecular regulatory networks involved in the fertilization process in Korean pine ovules through anatomical observation, physiological and biochemical assays, and transcriptome sequencing technology. The morphological and physiological results indicated that fertilization proceeds through the demise of the proteinaceous vacuole, egg cell division, and pollen tube elongation. Auxin, cytokinin, soluble sugar, and soluble starch contents begin to decline upon fertilization. Transcriptomic data analysis revealed a large number of differentially expressed genes at different times before and after fertilization. These genes were primarily involved in pathways associated with plant hormone signal transduction, protein processing in the endoplasmic reticulum, fructose metabolism, and mannose metabolism. The expression levels of several key genes were further confirmed by qRT-PCR. These findings represent an important step towards understanding the mechanisms underlying morphological changes in the Korean pine ovule during fertilization, and the physiological and transcriptional analyses lay a foundation for in-depth studies of the molecular regulatory network of the Korean pine fertilization process.

## 1. Introduction

Korean pine (*Pinus koraiensis*) is utilized for its wood and is also an important tree species of cold–temperate needle-leaved latitudinal forests [1]. Additionally, it is utilized for its medicinal properties, and its nuts are used in a variety of foods [2]. There are a low number of natural stands of Korean pine, and most plantations are relatively young (most of them have just started to set and have not entered the fruiting stage), making it difficult to meet the growing demand for its natural products [3]. Additionally, Korean pine’s female blossom buds only differentiate at the tip of the main trunk of the female tree, which takes approximately 15 months to progress from anthesis to seed maturation. It is, therefore, important to artificially regulate the reproductive growth stage of Korean pine to promote higher yields and quality [4]. Korean pine seeds can be forced to mature early by grafting to generate high seed setting rates, a method commonly used to propagate seedlings in forestry sectors [5]. Relatively few studies on Korean pine reproduction are available, and those that do exist are primarily focused on improving seed setting through cultivation measures such as fertilization and tree management. For example, the application of nitrogen fertilizer has been shown to promote seed setting and increase female blossom bud numbers. The application of GA_3_ has also been shown to significantly increase the number of female and male Korean pine blossom buds [6]. Research on Korean pine reproductive biology has mostly focused on issues such as ovule failure and poor fertilization, which hinder seed production [7]. Several genes, hormones, and other factors have been shown to impact the reproductive rate of Korean pine [8]. However, little work has been conducted aimed at understanding key genes involved in the regulation of Korean pine reproductive biology. 

All tree species of the *Pinus* genus have similar sexual reproductive processes. Studies on Chinese pine (*Pinus tabuliformis*) have indicated that the egg cell may serve as a suitable system for studying mRNA localization within cells of higher plants. During fertilization and early embryonic development, female gametocytes exist as multimeric structures [9], and all multimers are attached to microtubules (MTs) [10]. The phenomenon of sex reversal has also been observed in Chinese pine and Masson pine (*Pinus massoniana*), and the differentially expressed genes (DEGs) between normal branches and polycone branches were significantly up-regulated in sex-reversed polycone branches [11]. These studies shed light on the problems of the reproductive biology of Korean pine, including ovule abortion and poor fertilization. Therefore, it may be possible to solve the problem of low seed yield in Korean pine by finding DEGs at different stages of reproductive growth to artificially regulate fertilization or create new accessions.

In this study, we investigated the transcriptomes, anatomical structures, and the contents of soluble sugars, starch, IAA, ZR, and GA_3_ of Korean pine ovules before and after fertilization. We selected 10 DEGs in auxin signal transduction, cytokinin signal transduction, and mannose and carbohydrate metabolism pathways for qRT-PCR validation. This analysis revealed the morphological and physiological changes in Korean pine ovules during fertilization that were correlated with gene expression changes. The findings of this study can be used as a starting point to facilitate targeted breeding efforts aimed at accelerating the development of the Korean pine industry.

## 2. Results

### 2.1. Anatomical Observation of the Ovule Fertilization Process of Korean Pine

In the ovule two weeks before fertilization (ZA), the archegonial initial cell underwent a series of divisions to form the egg cell. The egg cell was large, surrounded by jacket cells and contained many protein vacuoles. The egg nucleus was located at the end of the micropyle, and the cytoplasm of the egg nucleus was granular. At this point, the pollen tube was not yet extended. In the ovule one week before fertilization (ZB), the protein vacuoles gradually disappeared and degraded, the cytoplasm became denser, and the egg nucleus’ volume increased. At fertilization (ZC), the pollen tube was partially elongated in one-half of the ovules, and the oocyte was close to the micropylar end, with an open channel at the micropylar end. Zygotic embryo cells were observed dividing immediately after fertilization in the remaining ovules. Since the instant of fertilization was not observed due to time and technical issues, ZC was considered to be the date closest to the instant of fertilization. One week after fertilization (ZD), the embryonic cells were observed developing (Figure 1). 

### 2.2. Physiological Analysis during Ovule Fertilization in Korean Pine 

#### 2.2.1. Changes in IAA and ZR Content

IAA content peaked at approximately 16.316 μg·g^−1^ fresh weight (FW) one week before fertilization, and then rapidly declined to a minimum of approximately 11.740 μg·g^−1^ FW one week after fertilization. ZR content increased two weeks before fertilization, began to decline one week before fertilization, and then significantly increased after fertilization (Figure 2).

#### 2.2.2. Changes in Soluble Sugar and Starch Contents

Soluble sugar and starch both showed similar trends during fertilization. Soluble sugar content increased significantly two weeks before fertilization and peaked the week before fertilization. The content was slightly lower at fertilization but increased continually after fertilization. Soluble starch content was slightly lower at all time points, and the difference was not significant (Figure 2).

### 2.3. RNA-Seq Analysis before and after Fertilization of Korean Pine Ovules

#### 2.3.1. Sequencing Data Evaluation and Statistical Results 

Transcriptome sequencing using the Illumina HiSeq high-throughput sequencing platform was performed on ovules two weeks before fertilization (ZA), one week before fertilization (ZB), at fertilization (ZC), and one week after fertilization (ZD). A total of 77.34 GB of clean reads were obtained after trimming and filtering out low-quality reads (Appendix A). By analyzing the error rate of sequencing quality control results, the percentages of Q20 bases were higher than 97%, the percentages of Q30 bases were higher than 93%, and the percentages of ambiguous bases N were all between 0.46% and 0.98% in the sequencing data used for assembly in the twelve samples. (Appendix A). These results indicated that the sequencing data were of high quality and suitable for downstream analyses.

The correlations between biological replicates were higher than correlations with other samples, indicating that the biological replicates were reliable (Figure 3).

The remaining high-quality sequences were aligned to the Korean pine reference genome (unpublished data) with alignment efficiencies ranging from 91.69% to 92.99%. The percentage of sequences aligned to unique locations in the reference genome was at least 62.35%. These results indicated that the selected reference genome was sufficient for downstream analysis and no significant contamination was present (see Appendix A for additional data).

#### 2.3.2. KEGG Pathway Analysis of DEGs among ZA, ZB, ZC, and ZD

In order to investigate the molecular events of the Korean pine fertilization process, we created Venn diagrams to determine the overlapping genes across all samples. There were 2048 unique DEGs between the ovules two weeks before fertilization and the ovule one week before fertilization, 5634 DEGs between the ovules one week before fertilization and the ovule at fertilization, and 3290 DEGs between the ovules at fertilization and the ovule one week after fertilization (Figure 4a). Next, cross-correlation analysis was performed on the DEGs at the four stages of fertilization. There were 7508 DEGs between the ovules two weeks before fertilization and the ovule one week before fertilization, of which 4376 DEGs were up-regulated and 3132 DEGs were down-regulated. A total of 13,404 DEGs were present between the ovule at one week before fertilization and the ovule at fertilization, of which 6523 were up-regulated and 6881 were down-regulated. A total of 9031 DEGs were present between the ovules at fertilization and the ovules one week after fertilization, of which 4805 were up-regulated and 4226 were down-regulated (Figure 4b). 

To further investigate the biological pathways involved in different stages of fertilization, we mapped DEGs to reference pathways in the KEGG database. Based on highly enriched DEG numbers, the top 20 most enriched pathways were determined. This analysis indicated that DEGs in the ZA vs. ZB comparison were annotated to 132 classical signal KEGG metabolic pathways, with significant enrichment seen in pathways associated with ribosomes, protein processing in endoplasmic reticulum, fructose and mannose metabolism, and proteasome (Figure 4c). The DEGs in the ZB vs. ZC comparison were annotated to 131 classical signal KEGG metabolic pathways, including protein processing and plant hormone signal transduction, and ubiquitin-mediated proteolysis (Figure 4d). The DEGs in the ZC vs. ZD comparison were annotated to 131 classical signal KEGG metabolic pathways, including the pentose phosphate pathway, carbon metabolism, biosynthesis of amino acids, steroid biosynthesis, and pyruvate metabolism (Figure 4e).

### 2.4. Gene Regulation during Ovule Fertilization of Korean Pine

#### 2.4.1. Genes Related to Plant Hormones

The early gene expression comparisons yielded significant enrichment in pathways associated with hormones, prompting us to investigate the role of hormones in Korean pine ovule formation further. The enriched pathways contained eight transduction pathways, including those associated with auxin (IAA), gibberellin (GA), abscisic acid (ABA), cytokinin (CTK), ethylene (ETH), brassinosteroids (BR), jasmonic acid (JA), and salicylic acid (SA). Since IAA and CTK signal transduction pathways were found to be the most enriched hormone pathways in the Korean pine ovule, we investigated them further. 

The binding of auxin to its receptor protein to degrade Aux/IAA has been shown to be necessary for auxin signal transduction [12]. Most of the Aux/IAA-related DEGs were up-regulated during the week of fertilization and the week after fertilization, and down-regulated during the two weeks before fertilization and the week before fertilization. This contrasted the expression changes seen in *AUX1*, suggesting that AUX/IAA negatively regulated the auxin signal transduction process. The GH3 and SAUR family DEGs reached their highest expression levels during the week of fertilization, and their lowest levels two weeks before fertilization (Figure 5a).

After CTK binds to its receptor CRE1, CRE1 is activated and transferred to AHP [13]. More AHP-related DEGs were up-regulated one week before fertilization and at fertilization, and all the DEGs except *BGI_novel_G014936* were down-regulated in the week after fertilization (Figure 5b). B-ARR-mediated networks contained more DEGs which were up-regulated during the week of fertilization and one week after fertilization, while genes associated with this network were down-regulated prior to fertilization. A-ARR-mediated networks contained genes which were significantly up-regulated during the two weeks before fertilization, all of which were down-regulated the week of fertilization. Taken together, these data indicate that down-regulation of A-ARR expression during the week of fertilization weakens the inhibition of B-ARR, leading to significant up-regulation of genes associated with B-ARR during the week of fertilization.

#### 2.4.2. Genes Related to Carbohydrate Metabolism

Differential expression testing and KEGG enrichment analysis indicated that genes associated with three major carbohydrate metabolic pathways were enriched, including genes associated with glycolytic and TCA pathways, ATP, NADH, and pyruvic acid. The expression levels of three rate-limiting enzymes (hexokinase 2.7.1.1, phosphofructokinase 2.7.1.11, and pyruvate kinase 2.7.1.40) in the first three time points were all low, which reduced the efficiency of glycolysis, inhibited growth and development, and decreased soluble sugar content (Figure 6a). All the above three enzymes were more highly expressed one week after fertilization, which accelerated glycolysis efficiency and provided energy for the growth and development of early embryos.

We also found that a large number of pyruvate dehydrogenase family genes may be involved in the regulation of the glycolytic pathway and TCA pathways. The expression profiles of DEGs were further analyzed via clustering. The E1-α (pyruvate dehydrogenase complex E1α subunit), E2 (dihydrolipoyl transacetylase), and E3 (dihydrolipoyl dehydrogenase) components of pyruvate dehydrogenase in PDHC (pyruvate dehydrogenase complex) were assessed in the ovules of Korean pine before and after fertilization. E1-α and E2 genes were down-regulated in the first three time points and significantly up-regulated one week after fertilization. However, a large number of E3 genes were up-regulated before fertilization and down-regulated after fertilization (Figure 6b).

KEGG enrichment analysis indicated that DEGs were enriched for the mannose metabolic pathway, including the key enzymes MAN and GMPP. Genes associated with both enzymes had very similar expression profiles. They were primarily down-regulated prior to fertilization, and then up-regulated one week after fertilization (Figure 6c).

#### 2.4.3. Genes Related to Heat Shock Proteins

Genes belonging to the HSP40, HSP70, and HSP90 families were all differentially expressed over the course of the time series. HSP40 and HSP70 genes were down-regulated two weeks before fertilization, but up-regulated one week before fertilization. HSP90 genes, on the other hand, were only up-regulated at two weeks prior to fertilization (Figure 7a).

#### 2.4.4. Genes Associated with SCF Complexes

The Skp1–Cul1–F-box protein (SCF complex) consists of SKP1, RBX1, CUL1, and F-box protein subunits. SKP1 is a key skeletal protein in the SCF complex that binds to both the F-box protein and Cul1. This complex has been shown to mediate the ubiquitination and degradation of different cyclins to control the cell cycle [14].

Two weeks before fertilization, ovules possessed central cells and protein vacuoles, which gradually disappeared as the cells became denser one week prior to fertilization. This process was accompanied by the nuclear division of the central cells and formation of the egg cell. The genes related to SKP1 were highly expressed in the week before fertilization, and the relative expression levels of most genes related to RBX1 were also high in the week before fertilization. Genes associated with SKP1 and RBX1 were all down-regulated during the week of fertilization and one week after fertilization (Figure 7b,c).

### 2.5. Verification of Genes Related to Fertilization Process by qRT-PCR

In order to verify the reliability of the transcriptome data, 10 genes related to fertilization were selected and the expression levels during ovule fertilization of Korean pine were analyzed by qRT-PCR. The real-time quantitative fluorescence detection results of the selected 10 differentially expressed genes were consistent with the change trend of the transcriptome sequencing results (Figure 8).

## 3. Discussion

### 3.1. Plant Hormone Signal Transduction Pathways

Plant hormones play an important role in several processes associated with the sexual reproduction of plants, including pollination and fertilization [15]. IAA has been shown to stimulate pollen germination and fertilization, and IAA content was positively correlated with fertilization rate. A low level of IAA has previously been shown to cause pollen tube deformity and hinder endosperm development before fertilization [16]. However, the appropriate auxin concentration causes activation of the Ca^2+^ pathway in the apical pollen tube under the influence of ATPase activity to promote pollen tube growth [17]. CTK impacts sperm–egg fusion and seed development, and high CTK content can enhance pollen viability and promote stigma recognition [18]. Therefore, it is critical to understand the molecular mechanism and expression patterns of genes associated with plant hormones during pollination and fertilization.

In our study, 93 DEGs associated with auxin signal transduction processes were identified, most of which were associated with ARF, SAUR, and Aux/IAA. IAA content was higher before fertilization, and excess IAA can promote the degradation of Aux/IAA to release ARF transcription factors and negatively regulate IAA (Figure 7a). Analysis of endogenous hormones showed that IAA levels decreased during the week of fertilization, and the transcriptome sequencing results showed that the AUX1 family genes were significantly down-regulated while the Aux/IAA, ARF, and GH3 family genes were significantly up-regulated during the week of fertilization. These results indicated that the AUX1 family genes played a positive regulatory role during fertilization, while the Aux/IAA, ARF, and GH3 family genes played a negative regulatory role. Moreover, a large number of genes in the SAUR family were up-regulated during fertilization, which was opposite to the results of IAA hormone content measurement, indicating that there was negative regulation of auxin synthesis and transportation. This finding is consistent with results indicating that the Os*SAUR39* gene plays a negative regulatory role in auxin synthesis and transportation in rice (*Oryza sativa*) [19]. In previous studies, some SAURs have been shown to be involved in auxin-regulated cell growth. For example, the *OsSAUR54* gene was found to be specifically expressed on the stigma of cotton (*Gossypium* spp.) to promote pollen tube development [20]. Polar auxin transport was also found to be important for the development of zygotes and embryos in *Arabidopsis thaliana* [21].

In this study, the most frequently annotated family of DEGs associated with CTK was B-ARR (24 DEGs). These genes had different expression patterns prior to and after fertilization in Korean pine. During fertilization, the expression levels of the B-ARR gene family were higher, while the expression levels of A-ARRs were lower. However, before fertilization, during egg cell mitosis, the expression level of A-ARR was the highest, while that of B-ARR was relatively low.

Studies have found that B-ARR genes can regulate the expression of cell cycle-related genes and affect the cell division process [22]. Our findings indicate that the highly expressed A-ARR family genes promote chromosome division in the egg cell, and negatively regulate the expression of B-ARR family genes to facilitate meiosis. Additionally, endogenous ZR content was higher one week before and after fertilization of Korean pine ovules. At these same time points, the expression levels of B-ARR family genes were significantly up-regulated, while the expression of A-ARR family genes was significantly down-regulated. These data indicate that during Korean pine fertilization, the B-ARR family genes positively regulate CTK content, while A-ARR family genes negatively regulate it [23]. All 10 A-ARRs in *A. thaliala* have been shown to be down-regulated in response to CTK signals, which is consistent with the findings of our study [24].

### 3.2. Carbohydrate Metabolic Pathway

The three rate-limiting enzymes related to the glycolytic pathway were all highly expressed one week after fertilization, resulting in an increased rate of glycolysis that provides energy for the growth and development of early embryos. It has previously been shown that pyruvate kinase is expressed from the beginning of seed embryogenesis, and both pyruvate kinase and exogenous NADPH promote the growth and development of seeds [25].

The processes of plant seed growth and development are usually accompanied by the accumulation of lipids. The decomposition of pyruvic acid provides energy for lipid anabolism, and its decomposition products are the precursors of lipid biosynthesis [26]. We found that low levels of E1-α and E2 in the early stage of fertilization promoted fertilization and increased the synthesis of oil and fat in the seed after fertilization. A large number of E3 genes were up-regulated before fertilization and down-regulated after fertilization, indicating that the up-regulated expression of E3 could promote fertilization and improve the fertilization rate. Suppression of E1-α subunit activity in the tapetal cells of sugar beet (*Beta vulgaris*) anthers has been shown to lead to microspore exine dysplasia and male sterility [27]. When E3 activity was inhibited in hamsters (*Cricetinae*), the sperm fertilization rate was significantly reduced, indicating the importance of E3 in the fertility of hamster sperm. These findings in animals are also consistent with our results in Korean pine [28].

The prerequisite for fertilization is to attract the pollen tube and send the sperm into the female gametophyte through the pollen tube extension, a process that needs to be precisely regulated [29,30]. In *A. thaliana*, *PEANUT1* (*PNT1*) is a homologue of the mammalian mannosyltransferase PIG-M, and the *pnt1* mutant displays pollen viability and embryonic development defects [31]. These results indicate that plant and mammalian homologs of PIG-M play similar roles in reproduction. Studies have also shown that GDP-mannose is involved in the synthesis of carbohydrates in structural components of the plant cell wall [32]. Studies on the double fertilization process in rice showed that the cell wall might be damaged by the growth of the pollen tube through the cell later, which is necessary for successful fertilization. The degradation of the cell wall also provides energy for the growth of the pollen tube. The enzymes involved in the mannose pathway were mainly MAN and GMPP, and we found that 14 genes related to these two catalytic enzymes were up-regulated in the pollen tube elongation stage. The resulting increase in mannose content was conducive to the degradation process of the cell wall and provided energy for pollen tube elongation [33]. Based on our analysis of soluble sugar and starch contents during fertilization, we propose that their changes were primarily due to changes in the expression of MAN and GMPP enzymes. The up-regulation of these two enzymes and the increase in soluble sugar and starch contents, therefore, enhanced pollen tube extension.

### 3.3. Heat Shock Proteins

There were a large number of protein vacuoles in the archegonium of Korean pine two weeks before fertilization, but they disappeared one week before fertilization when the cytoplasm became denser. Based on our results, it is likely that the disappearance of protein vacuoles is part of programmed cell death (PCD), which is impacted by HSP40 and HSP70.

The heat shock reaction is employed by organisms to survive a variety of environmental stresses. HSPs are a class of molecular chaperones with highly conserved amino acid sequences and functions. Several biological processes are affected by HSP expression, including cell proliferation and differentiation. Based on their molecular weight and amino acid sequence homology and function, HSPs are divided into six types, including HSP100, HSP90, HSP70, HSP60, HSP40, and small-molecule HSPs [34]. In our study, three HSP families contained genes that were differentially expressed during the time series, including HSP40, HSP70, and HSP90. A large number of up-regulated genes were expressed in the two weeks before fertilization compared with the week prior to fertilization, suggesting that HSP40 and HSP70 families may be involved in the PCD process associated with plant embryonic development. Proteomic studies during the development of zygotic embryos in the Brazilian pine (Araucaria angustifolia) have also revealed massive accumulations of HSPs in globular embryos during late embryonic development [35]. Similarly, the involvement of HSP70 in the regulation of PCD has been demonstrated in rice [36]. In a recent study of leaf development in Aponogeton madagascariensis, HSP70 content was found to be highest before and during PCD. Reducing the content of Hsp70 inhibits the induction of PCD, indicating that it plays a role in PCD during leaf development [37].

Overall, a considerable number of up-regulated HSP40 and HSP70 family protein-coding genes were identified in the two weeks before fertilization and one week before fertilization of Korean pine. The peak expression of these families coincided with the disappearance of protein vacuoles. Based on these findings, HSP40 and HSP70 appear to play key regulatory roles in the PCD process and in the disappearance of protein vacuoles in Korean pine.

### 3.4. SCF Complex

Previous studies have shown that the SCF complex plays an important role in the development of male and female gametophytes, as well as in early embryonic development [38]. Research in wheat (*Triticum aestivum*) has revealed that reductions in the level of *SKP1* affect the growth and development of pollen grains, leading to premature death and male infertility [39]. In *A. thaliana*, the mutation of *ASK*, a homolog of *SKP1*, affects cell division, leading to developmental delay and seed death [40]. Our findings that *SKP1*-related genes were up-regulated during egg cell division and down-regulated following fertilization are consistent with previous *SKP1* research in other systems. Abnormal expression of SKP1 proteins in both plants and animals has been shown to cause embryogenesis abnormalities, and overexpression of *SKP1* blocks the development of early embryos in mice (*Mus musculus*) [41].

The role of *RBX1* in plants has not been studied sufficiently, but one report on Arabidopsis indicated that it is involved in the formation of lateral roots [42]. In animals, low expression of the *RBX1* gene was also shown to be embryo-lethal in *Drosophila* (*Drosophilidae*) and mice [43,44]. The expression changes seen in genes related to *RBX1* in our study further support its role in development, and indicate that *SKP1* and *RBX1* may work in concert to promote the formation and expansion of the egg cell in Korean pine. Lower expression of these genes may, therefore, block embryo development.

## 4. Materials and Methods

### 4.1. Plant Materials

The Korean pine ovules were obtained from the Hongwei seed garden, Baishan City, Jilin Province, China (127°27′32″ E, 42°13′36″ N), family 425. The study was performed on elite clonal grafted mother trees colonized in 1989. The collection was performed in June of the second year (2021) after natural pollination. The samples for the two weeks before fertilization were obtained on June 8 (ZA), while the samples for the one week before fertilization were obtained on June 15 (ZB). The samples for the week of fertilization were obtained on June 22 (ZC), and the samples for the one week after fertilization were obtained on June 30 (ZD). Samples were sealed in plastic bags with ice packs and sent to a laboratory after collection. The ovules were separated from the ovule scales using forceps and dissecting needles. The ovule sample was divided for further usage. All ovules were divided equally in triplicate. A portion was put into the FAA fixative solution, followed by air evacuation and placement in a 4 °C freezer for paraffin section production. Another portion was frozen in liquid nitrogen and stored at −80 °C in a freezer for RNA extraction and determination of hormone content. The remaining portion was dried and used for determination of soluble sugar and starch contents.

### 4.2. Anatomical Observation of Korean Pine Ovules

The materials in the FAA stationary liquid were fixed into permanent sections using the classical paraffin sectioning method, with slight modifications described by Li et al. [45]. The sections were then stained with safranin–fast green and observed under a fluorescence inverted microscope (Zeiss Axio Observer, Oberkochen, Germany).

### 4.3. Measurement of Physiological Indexes of Korean Pine Ovules

Measurement of enzymes associated with plant hormones were carried out using a linked immunosorbent assay (ELISA). The kit was provided by the crop control laboratory of China Agricultural University, and the contents of indoleacetic acid (IAA) and cytokinin (CTK) were detected using a microplate analyzer (Infinite F50). Phytohormone extraction assay methods were performed as described in the kit instructions, with 3 biological replicates per sample. The oven-dried samples were assayed for soluble sugar and starch contents using the anthrone–sulfuric acid method, with three biological replicates per sample [46].

### 4.4. Extraction of RNA from Korean Pine Ovules and Transcriptome Sequencing

Total RNA of the ovules frozen in liquid nitrogen and stored at −80 °C was extracted. RNA integrity and quality were assessed using an Agilent 2100 Bioanalyzer (Agilent) and a NanoDrop 2000 ultra-micro-spectrophotometer (Thermo Scientific, Waltham, MA, USA). RNA library construction and Illumina sequencing (Illumina, San Diego, CA, USA) were then performed by BGI Inc. (Shenzhen, China).

### 4.5. RNA-Seq and Transcriptome Data Processing

The raw data were filtered using SOAPnuke (v1.4.0), which was developed by the BGI School of Life Sciences, Shenzhen, China. Reads containing junction contamination, reads with an unknown base N content greater than 5%, and reads of low quality (more than 20% of the bases having a quality score less than 15) were removed to obtain clean reads. Unigenes were annotated using BLAST software for KEGG annotation.

### 4.6. Analysis of Differentially Expressed Genes

Based on the KEGG annotation results as well as the official classifications, we subjected the DEGs to functional classification. Enrichment analysis was performed using the R package Phyper. The *p*-values were calculated as follows:
$$P=1-\sum_{i=o}^{m-1}\frac{\left(\begin{array}{c} M\\ i \end{array}\right)\left(\begin{array}{c} N-M\\ n-i \end{array}\right)}{\left(\begin{array}{c} N\\ n \end{array}\right)}$$

The *p*-values were then FDR-corrected, and tests with Q-value ≤ 0.05 were considered significantly significant.

Based on the KEGG annotation results as well as the official classification, we classified the DEGs into biological pathways while performing enrichment analysis using the Phyper function in R software. The *p*-values were calculated as above. The *p*-values were then FDR-corrected, and tests with Q-value ≤ 0.05 were considered significantly enriched.

### 4.7. Quantitative PCR (qRT-PCR) Analysis

Ten randomly selected DEs (Appendix A) in the auxin signaling, CTK signaling, fructose and mannose metabolism, and carbohydrate metabolism pathways were subjected to qPCR validation. The ovules two weeks before fertilization (ZA), one week before fertilization (ZB), at fertilization week (ZC), and one week after fertilization (ZD) were used for RNA extraction and then reverse transcribed to cDNA as a template. Each sample included three biological replicates, with 18S and c154707.graph_c1 used as the two internal reference genes. SYBR Green I was used to detect the expression of target genes in the samples. RNA OD was measured on a ScanDrop 100 ultra-micro-nucleic acid protein assay instrument (Analytik Jena, Jena, Germany) using a reversed first-stand cDNA synthesis kit (Applied Biosystems, Aidelab, Germany) in a 20 μL reaction system. Detection of fluorometric analysis was accomplished on an Analytik Jena-qTOWER 2.2 fluorescence quantitative PCR instrument (Applied Biosystems, Waltham, MA, USA). Relative gene expression levels in each sample and each group were calculated using 2 ^−△△^CT.

## 5. Conclusions

In this study, the ovules (ZA) two weeks before fertilization, the ovule (ZB) one week before fertilization, the ovule (ZC) one week after fertilization, and the ovule (ZD) one week after fertilization were taken as the research objects, and morphological observation, physiological and biochemical analysis, as well as transcriptome sequencing analysis were performed to comprehensively explore the influencing factors of the ovule fertilization process. The results showed that the process of fertilization included the disappearance of protein vesicles, the division of oocytes, and the extension of pollen tubes until fertilization, and the measurement results of the content of various physiological indicators during the fertilization week showed a decreasing trend to different degrees. At the same time, transcriptome sequencing results showed that the differential genes were mainly involved in plant signal transduction, carbohydrate metabolism, endoplasmic reticulum protein processing, and the ribosome pathway. 

On this basis, we found that the IAA content of the ovule during perifertilization was decreased, when the AUX1 family genes were significantly down-regulated and played a positive regulatory role. B-ARR family genes positively regulate cytokinin content and act on pollen tube extension and cell division. The down-regulated expression of E1-α and E2 components of pyruvate dehydrogenase and the up-regulated expression of the E3 component were conducive to the smooth progress of fertilization before fertilization itself. MAN and GMPP, two catalytic enzymes of the mannose pathway, mainly acted on the extension process of the pollen tube and were consistent with the changes in soluble sugar and starch contents. A large number of up-regulated HSP40 and HSP70 family protein-coding genes in the ZB stage regulate the PCD process of protein bubble disappearance. SKP1 and RBX1 family genes in SCF complexes are mainly up-regulated before fertilization, which can promote oocyte division.

## Figures and Tables

**Figure 1 ijms-24-06570-f001:**
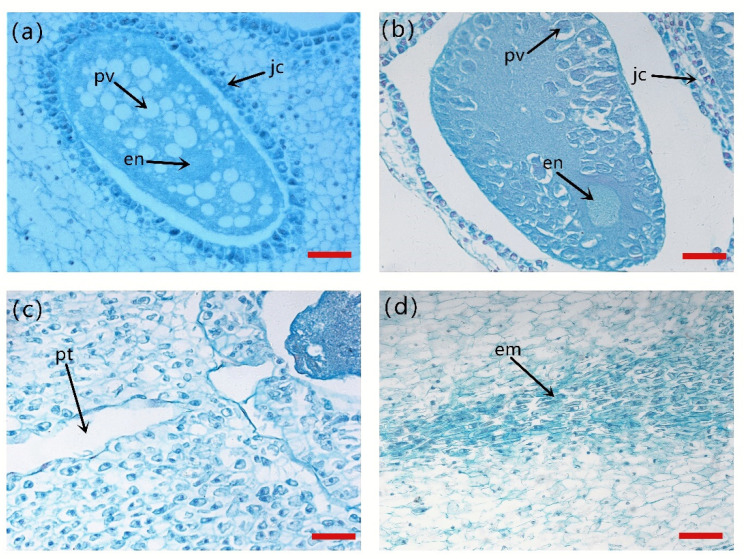
Anatomical observation on ovule fertilization of Korean pine. (**a**) Ovule of Korean pine two weeks before fertilization, bar = 5 μm; (**b**) Ovule of Korean pine one week before fertilization, bar = 5 μm; (**c**) Ovule of Korean pine at fertilization week, bar = 5 μm; (**d**) Ovule of Korean pine one week after fertilization, bar = 10 μm.

**Figure 2 ijms-24-06570-f002:**
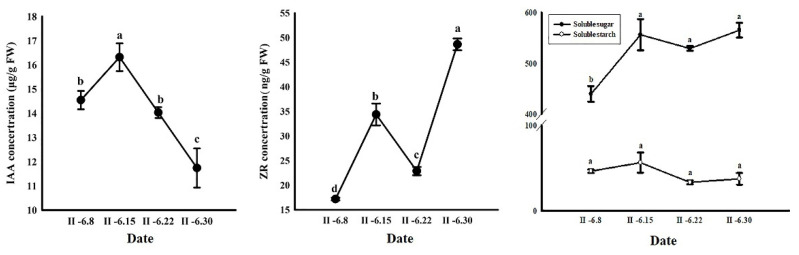
IAA, ZR, soluble sugar, and starch contents in Korean pine ovules before and after fertilization. Error bars indicate the SD of three biological replicates. Lowercase letters represent significant differences at *p* < 0.05.

**Figure 3 ijms-24-06570-f003:**
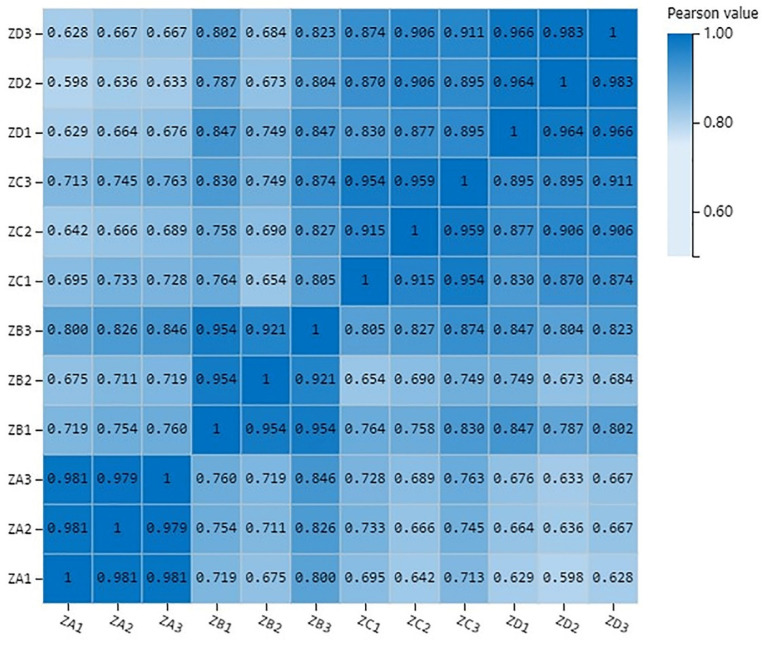
Biological repeat correlation of ovule sample of Korean pine.

**Figure 4 ijms-24-06570-f004:**
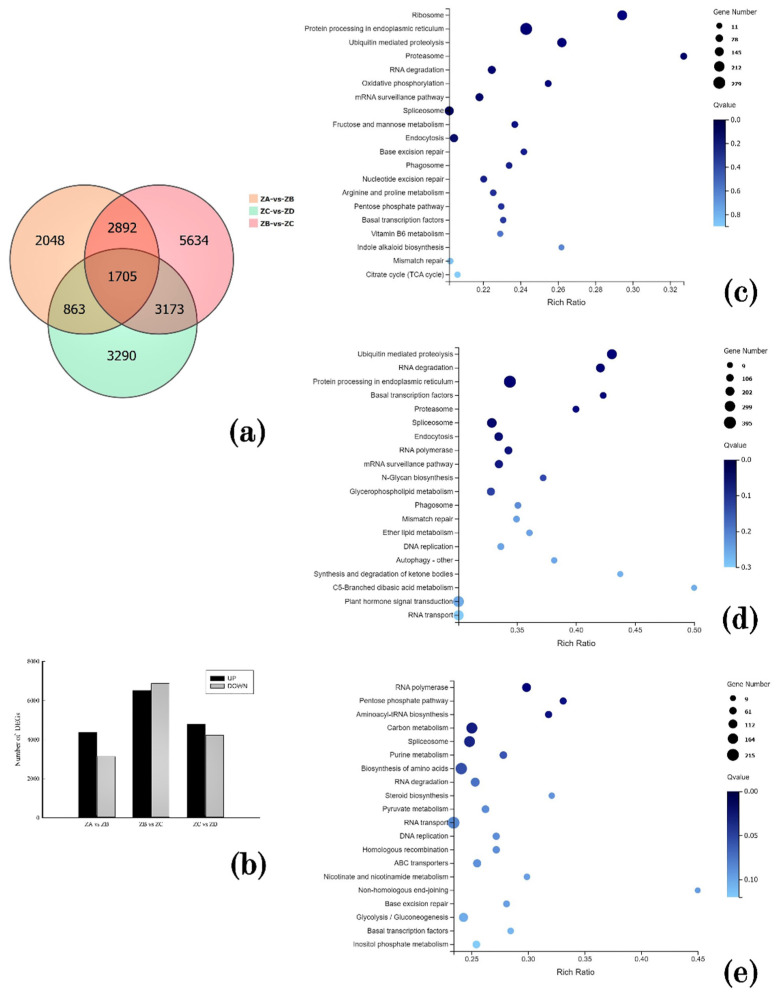
(**a**) Venn diagram of data of DEGs; (**b**) Statistical diagram of data of DEGs. (**c**) Statistical Diagram of Annotated Classification of DEGs of ZA vs. ZB. (**d**) Statistical Diagram of Annotated Classification of DEGs of ZB vs. ZC. (**e**) Statistical Diagram of Annotated Classification of DEGs of ZC vs. ZD.

**Figure 5 ijms-24-06570-f005:**
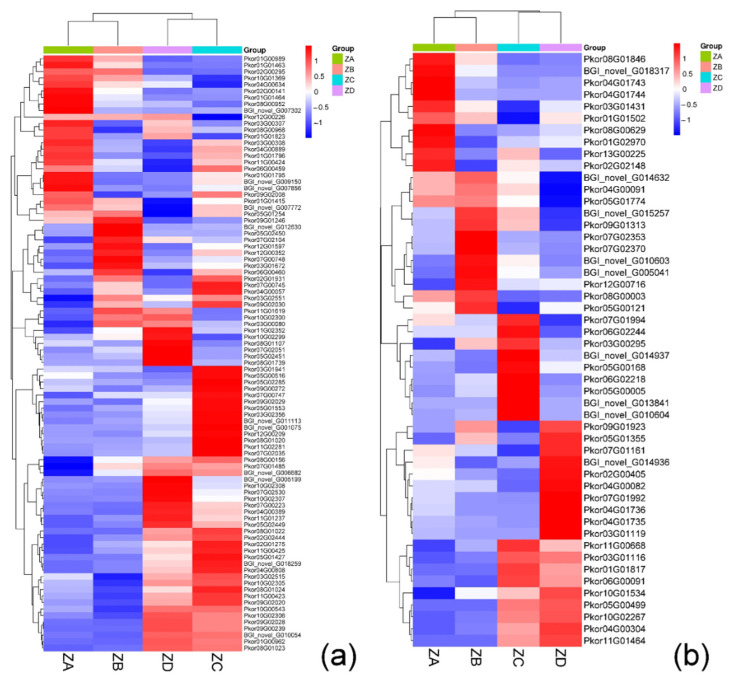
(**a**) Expression of auxin-related differential genes in Korean pine; (**b**) Expression of cytokinin-related differential genes in Korean pine.

**Figure 6 ijms-24-06570-f006:**
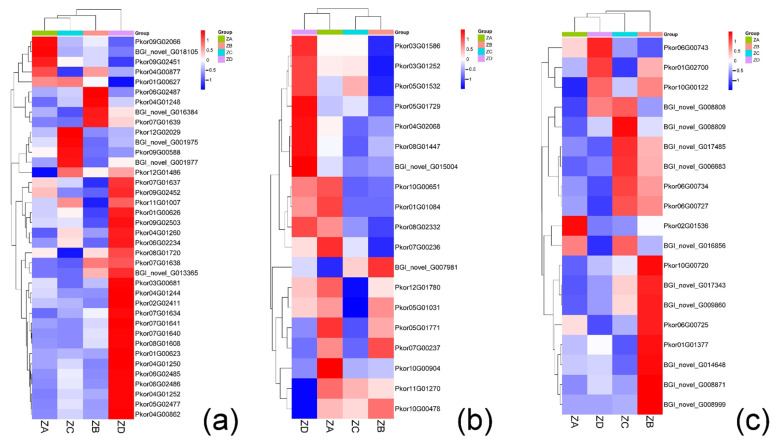
(**a**) Expression of differential genes related to carbohydrate metabolic pathways in Korean pine before and after fertilization; (**b**) The expression of differential genes related to pyruvate dehydrogenase before and after fertilization of Korean pine; (**c**) Differential gene expression related to mannose in Korean pine before and after fertilization.

**Figure 7 ijms-24-06570-f007:**
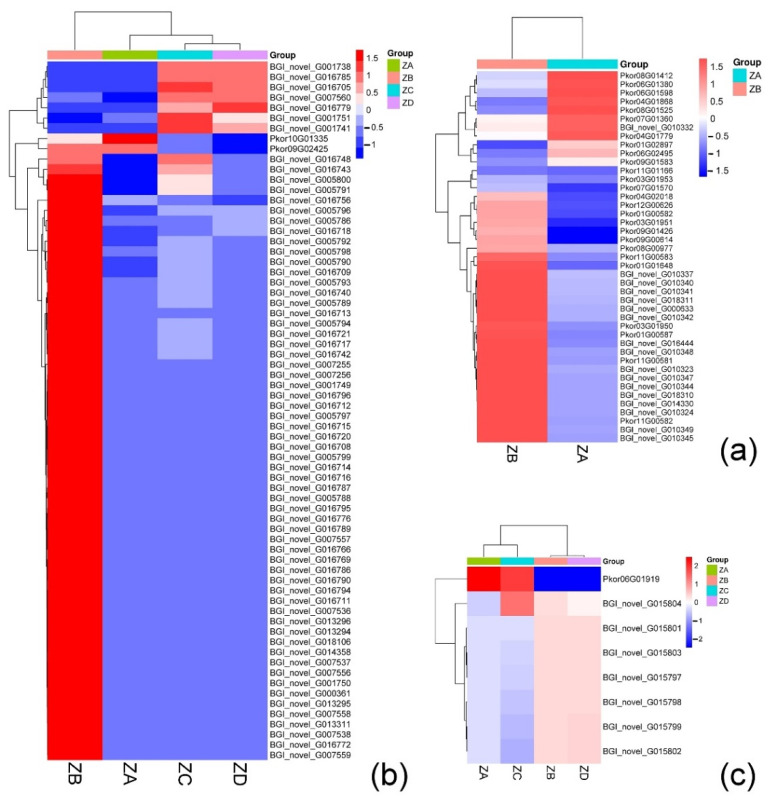
(**a**) Expression of differential genes related to heat shock proteins before and after fertilization of Korean pine; (**b**) Expression of differential genes related to SKP1 in Korean pine before and after fertilization; (**c**) Expression of RBX1-related differential genes in Korean pine before and after fertilization.

**Figure 8 ijms-24-06570-f008:**
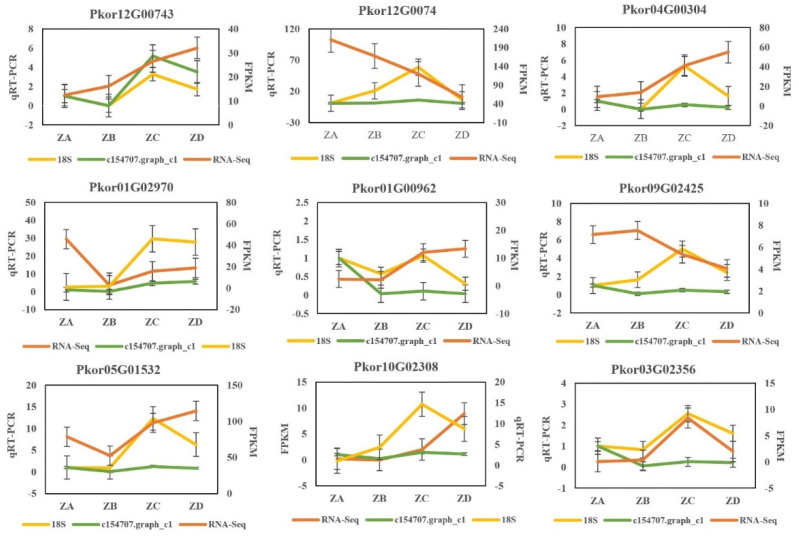
Verification results of 10 differential genes qRT-PCR before and after ovule fertilization of Korean pine.

## Data Availability

All RNA-seq reads were deposited at NCBI (Bioproject ID: SUB12731425).

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
