# Peer review of "Transcriptomic Analysis of Hormone Signal Transduction, Carbohydrate Metabolism, Heat Shock Proteins, and SCF Complexes before and after Fertilization of Korean Pine Ovules"

_ijms, 2023, doi:10.3390/ijms24076570_

Round 1

Reviewer 1 Report

The paper “Transcriptomic analysis of hormone signal transduction, carbohydrate metabolism, heat shock proteins, and SCF complexes before and after fertilization of Korean pine ovules” by Xiaoqian Yu, Xueqing Liu, Yuanxing Wang, Yue Zhang, Hailong Shen and Ling Yang describes a thorough study of Korean pine fertilization process. Unlike European pines that “saves” the pollen in dormancy for the next year and it would be hard to tell the exact time of fertilization in advance, P. koraiensis was observed for 4 weeks that covered 2 weeks before fertilization, the week when fertilization was supposed to occur and the week after fertilization.

Structural changes were found via anatomic observation and reported, 4 types of analytes were measured: IAA, ZR, soluble sugar and starch contents. Gene expression was studied, with focus to plant hormones, carbohydrate metabolism, heat shock proteins and SCF complexes. Concentration changes of other metabolites are not reported, but gene expression comparisons are detailed. A group of 10 genes was selected for the verification by qRT-PCR. The findings support the hypothesis nicely, as explained in a detailed discussion, and are in good accordance with current state of knowledge about other species.

The paper is written in good, simple scientific English, well organized, the amount of figures is appropriate and illustrative. The results built the base for understanding the studied process and while the authors don’t exactly suggest the way to increase P. koraiensis fertility, they successfully identified key factors that influence and regulate the process and paved the way to potential solution. Crops fertility is a topic of interest for many readers, and the manuscript is thus recommended to be accepted to the IJMS journal.

Small advice to fix:

Line 5: … Hailong Shen1, 3**, Ling Yang1 and 3**: this seems to the reader as if Hailong Shen had two affiliation (1, 3), Ling Yang had one (1) and there was a missing author from 3**

think about changing it to “… Hailong Shen1, 3** and Ling Yang1,3**”

Line 404: … the ovule was divided into three parts. As the ovule usually consist of three parts, integument, nucellus and gametophyte, the reader will expect some cutting and separation here. Supposedly, it was the ovule sample that was divided for further usage, and you might want to change that sentence.

These are very minor issues and can be revised quickly.

You might also want to get a cooperation with someone on the HPLC side of metabolomics for your next paper. Concentration changes of hormones and antioxidants would be interesting, and gemmotherapy is also rather untapped corner. Good luck with your further research.

Author Response

Dear Reviewer,

Our sincere thanks to you for the time and effort that you have put into reviewing our manuscript! We found all the comments very constructive and helpful, and have revised our manuscript according to all comments. Please find, below, our point-by-point response to the comments raised.

Thank you for considering our revised manuscript!

Point1:Line 5: … Hailong Shen1, 3**, Ling Yang1 and 3**: this seems to the reader as if Hailong Shen had two affiliation (1, 3), Ling Yang had one (1) and there was a missing author from 3**

think about changing it to “… Hailong Shen1, 3** and Ling Yang1,3**”

Response 1: Has been modified to: Hailong Shen3**, Ling Yang1**

Point2: Line 404: … the ovule was divided into three parts. As the ovule usually consist of three parts, integument, nucellus and gametophyte, the reader will expect some cutting and separation here. Supposedly, it was the ovule sample that was divided for further usage, and you might want to change that sentence.

Response 1: Has been modified to: The ovule sample was divided for further usage. All ovules were divided equally into triplicate.

Reviewer 2 Report

Authors investigated the fertilization process in Korean pine ovules using a combination of anatomical, physiological, biochemical, and transcriptomic analyses. They reported specific features associated with the fertilization and revealed a set of differentially expressed genes before and after fertilization. These genes were analyzed according to their previously known functions in signaling pathways, and some of them were checked using the qRT-PCR analysis. Overall, the study is well justified, and the results are solid, leading to reasonable conclusions. In the Discussion section, the authors provide a quite extensive comparison of their results to the current knowledge in a wider context. Therefore, I do not have any essential questions or comments to the study. The only technical remark concerns the representation of the material: Figures 2, 4, 6, 7, and 10 have a too low resolution. It is hard (and sometimes impossible) to distinguish marks and captions on the axes.

Author Response

Dear Reviewer,

Our sincere thanks to you for the time and effort that you have put into reviewing our manuscript! We found all the comments very constructive and helpful, and have revised our manuscript according to all comments. Please find, below, our point-by-point response to the comments raised.

Thank you for considering our revised manuscript!

Point: Figures 2, 4, 6, 7, and 10 have a too low resolution.

Response 1: We have improved the resolution of the pictures mentioned above.